# Environmental Quality bOX (EQ-OX): A Portable Device Embedding Low-Cost Sensors Tailored for Comprehensive Indoor Environmental Quality Monitoring

**DOI:** 10.3390/s24072176

**Published:** 2024-03-28

**Authors:** Jacopo Corona, Stefano Tondini, Duccio Gallichi Nottiani, Riccardo Scilla, Andrea Gambaro, Wilmer Pasut, Francesco Babich, Roberto Lollini

**Affiliations:** 1Institute for Renewable Energy, Eurac Research, 39100 Bolzano, Italy; 2Center for Sensing Solutions, Eurac Research, 39100 Bolzano, Italyriccardo.scilla@gmail.com (R.S.); 3Photonics Integration, Electrical Engineering Department, Eindhoven University of Technology, 5600 MB Eindhoven, The Netherlands; 4Environmental Sciences, Informatics and Statistics Department, University Ca’ Foscari, 30172 Venezia, Italygambaro@unive.it (A.G.);; 5Dipartimento di Ingegneria e Architettura, Università di Parma, 43124 Parma, Italy; 6College of Engineering, University of Korea, Seoul 06591, Republic of Korea

**Keywords:** indoor environmental quality monitoring, IoT sensing, low-cost sensors, IEQ field measurements, calibration methods

## Abstract

The continuous monitoring of indoor environmental quality (IEQ) plays a crucial role in improving our understanding of the prominent parameters affecting building users’ health and perception of their environment. In field studies, indoor environment monitoring often does not go beyond the assessment of air temperature, relative humidity, and CO_2_ concentration, lacking consideration of other important parameters due to budget constraints and the complexity of multi-dimensional signal analyses. In this paper, we introduce the Environmental Quality bOX (EQ-OX) system, which was designed for the simultaneous monitoring of quantities of some of the main IEQs with a low level of uncertainty and an affordable cost. Up to 15 parameters can be acquired at a time. The system embeds only low-cost sensors (LCSs) within a compact case, enabling vast-scale monitoring campaigns in residential and office buildings. The results of our laboratory and field tests show that most of the selected LCSs can match the accuracy required for indoor campaigns. A lightweight data processing algorithm has been used for the benchmark. Our intent is to estimate the correlation achievable between the detected quantities and reference measurements when a linear correction is applied. Such an approach allows for a preliminary assessment of which LCSs are the most suitable for a cost-effective IEQ monitoring system.

## 1. Introduction

Back in the 1990s, according to [1], people in the U.S. used to spend almost 87% of their time in closed environments (houses, offices, schools, etc.). This percentage recently increased due to the change in people’s habits ascribable to the COVID-19 pandemic [2,3] since people in many countries were forced to work from home. With this background in mind, now more than ever, it is key to assess indoor environmental quality (thermal, visual, acoustic, and air quality) to better understand the complex interaction between building systems, indoor conditions, and users.

The presence of different pollutants in indoor environments poses several questions about the health effects of the environments people live in. Household air pollution is ranked as the ninth largest global burden of disease risk [4]. Despite being more urgent in some areas of the planet, these issues are shared, to some extent, by most of the world’s population. There is a pressing need to understand the interactions between outdoor contaminants, indoor sources, building envelopes, ventilation systems, and users. Air quality is just one of the four parameters constituting IEQ that is now in the spotlight due to the pandemic. Indoor air quality (IAQ) refers to the quality of the air inside a building, i.e., pollutants and particle concentrations. It is highly relevant to the health of human beings [5]. Indoor environmental quality, or IEQ, affects a broader range of factors, like visual, acoustic, and hygrothermal comfort. It not only affects our physical health but also our mental well-being and productivity [6]. In addition to the potential health implications of air contaminants (see [7]), it is clear that there is much to investigate concerning contaminant concentrations’ potential impacts on human cognitive performance [8,9]. There has been much research on other parameters contributing to the IEQ. Even though it has been studied for a long time, thermal comfort seems to need more research when it comes to indoor parameters and human interactions. Two recent studies dealing with indoor temperatures [10] and cognitive performance [11] point in opposite directions, underlining the need for deeper insight into this topic.

In recent years, the growing availability of low-cost sensors (LCSs) able to track several environmental parameters has triggered the successful exploitation of monitoring devices in indoor environments. Many companies started producing and selling LCSs to monitor IEQ through either hygrothermal parameters (i.e., air temperature, globe temperature, surface temperature, air velocity, and relative humidity), visual and acoustic comfort, or other indoor air quality (IAQ) parameters, such as harmful gas concentrations and the presence of particulate matter (PM).

Researchers can better understand the monitored environment and building systems’ performance as they are now able to simultaneously measure many parameters at a reasonable cost. Still, to obtain reliable results, the accuracy and uncertainty of the measurements must be known and set beforehand.

The possibility of creating a diffused network of sensors, connected to the building management and control system (BMCS), can lead to a drastic improvement in IEQ. Several government-funded projects and policies promoted by different institutions [12] are currently supporting research on LCSs for IEQ. The modern paradigm of nearly/net zero energy buildings (nZEBs)—which were made mandatory for new construction projects in the EU in January 2021 (see [13])—requires a network of sensors to monitor in real time the indoor environment, to reduce energy consumption, and to increase occupants’ comfort. Electricity consumption has been increasing from year to year in private blocks, in apartments, in public institutions, and in data centers, as shown in [14]. Due to the increasing use of all types of electrical devices, in the field of control systems for human health, saving electrical energy has become a very important feature for monitoring devices.

Ref. [15] describes a paradigm shift from the early development of expensive, stationary, and complex monitoring instruments employed by the few able to afford them—i.e., governments, industry, and researchers—to the widespread use of low-cost, portable, and user-friendly systems that can be directly owned by communities and individuals. Since then, many studies have reported on LCS-based monitoring stations [16,17,18,19,20,21]. On the one hand, this allowed for the rise of a new concept of “smart” sensors, which are intrinsically cheaper, lighter, smaller, and easily connected to the Internet. On the other hand, these platforms happen to still be poorly characterized in terms of sensors’ uncertainty and/or project-specific tools with a low level of adaptability. Nevertheless, the availability of these sensors has led to many scientific publications, and over the last decade, several reviews have been published. Each of these reviews is focused on a different topic, some examples of which are listed below. Ref. [12] analyzed the impact of sensor technology in different communities and classified different projects according to the nature of their funding, either publically funded, commercially funded, and/or crowdfunded. The authors reported that around 30% of the projects turned out to be of the latter type, i.e., funded by a private company or by a group of people via crowdfunding. According to the authors, this represents a paradigm shift in air quality monitoring, which has been historically implemented by government organizations. Other works have focused on characterizations of one group of sensors/monitors, such as for the monitoring of PM [22,23] or gaseous pollutants [24,25,26]. Other studies present a general state-of-the-art overview and relevant applications of LCSs for IEQ [27,28,29,30,31,32,33]. Since these studies analyzed different databases and had different search queries and selection criteria, they often did not obtain the same results. Ref. [32] detected an overall lack of accuracy in the measured environmental parameters reported in several works. This was partly due to the great number of studies on the topic that can be considered “grey literature”, which did not use or detail a technical approach in terms of the calibration and performance of sensors. Moreover, even in the scientifically validated literature, the authors found that almost 40% of the analyzed works did not calibrate the sensors used or validate their data. According to the authors, this may hamper data reliability and eventually reduce people’s estimations of the beneficial effects of introducing such sensors into their everyday lives.

### 1.1. Literature on LCSs

The literature on LCSs is quite vast. In this section, we report the papers that are relevant to this work. Ref. [34] tried to address some of the most common issues concerning LCS-based IEQ monitoring platforms (i.e., the trade-off between energy consumption, measurement accuracy, and costs). The authors also developed algorithms to smooth out the collected data, carry out auto-calibration when needed, and optimize data transmission to save energy. No proper calibration was performed on the sensors. Another extensive series of studies was completed at the University of Sydney, Australia [20,21,35]. These papers introduce the SAMBA (Sentient Ambient Monitoring of Buildings in Australia) system, designed for monitoring air quality and thermal, visual, and acoustic comfort in office buildings. For all the measured quantities, the authors listed the main reference standards in different countries. Also, they provided the results of comparison tests carried out with calibrated laboratory equipment, thus showing the possible use of LCSs for IEQ. A similar concept was developed in [19,36] using the nEMoS (nano Environmental Monitoring System) device, equipped with CO_2_, illuminance, air speed, air temperature, radiant temperature, and relative humidity sensors. The system featured an open-source hardware and software platform to handle and share the measured data. Some of the sensors were compared with calibrated laboratory equipment to verify their behavior. The results of the work of [37] took advantage of the Arduino platform and Xbee module for data transmission. The authors claimed to have performed a proper calibration of the gas sensors by comparing their readings with a professional-grade instrument inside a sealed bag, in which different concentrations of several gases were introduced. In the same year, the authors of [16] published the results of the OSBSS (Open Source Building Science Sensors) system, which uses open-source software to collect data on air temperature, relative humidity, illuminance, motion, CO_2_, and pressure. The system had no wireless data transmission module, but it could acquire long-term data and store them on a local drive. The system was powered by a lithium-ion cell with an expected life of 1 year with a 1 min logging interval. Ref. [38] presented a different solution. In the framework of the AirSenseEUR project, the authors developed an integrated system to monitor air temperature, relative humidity, pressure, and the concentration of O_3_, NO_2_, NOx, and CO. When integrated into an IoT (Internet of Things) network, the monitoring platform, connected to the BMS, can thoroughly assess (and affect) the thermal comfort and air quality in buildings. However, the authors presented no details about the sensors’ performances, though they described future steps toward proper sensor calibration. An interesting option is presented by the authors of [17], since their battery-powered LCL (Low-Cost Logger) capable of measuring air temperature, air speed, mean radiant temperature, relative humidity, illuminance, sound pressure level, and CO_2_ concentration also features a graphic user interface to collect information about the personal comfort of occupants through a survey presented on the touch screen of the device. Still, no details about the sensors’ performance in terms of measurement accuracy were reported, except the ones provided by the sensors’ manufacturers. Regarding sensor calibration, the authors of [39] developed a procedure to comparatively cross-calibrate their low-cost sensors (monitoring air temperature, relative humidity, CO_2_, VOC, HCHO, NO_2_, and O_3_) by synchronizing their signals after simultaneous data acquisition. However, this procedure did not appear to be reliable in providing the actual sensors’ accuracy. Ref. [40] designed a system based on Arduino’s system to monitor temperature, relative humidity, illuminance, CO_2_, VOC, PM2.5, and room occupancy. The system was tested in the field and provided long-range data acquisition. As in some of the previously mentioned studies, no information about sensor calibration was provided, since the sensors were assumed to be calibrated by the manufacturer. Ref. [41] focused on the aspects related to data transmission protocols, intending to reduce power consumption related to data transmission for a monitoring system capable of collecting data about temperature, relative humidity, PM, gas concentration (O_2_, O_3_, CO_2_, CO, NO_2_, etc.), illuminance, electromagnetic radiation, and motion. No detailed information about the sensors’ accuracy was reported in this study either. A simple but effective system was presented by the authors of [42]. They developed an open-source system based on Raspberry Pi, capable of monitoring air temperature, globe temperature, relative humidity, pressure, CO_2_, VOC, PM, illuminance, airspeed, and sound pressure level. Its wooden case, designed to avoid system overheating and to allow for the collection of thermal comfort surveys from occupants, is the most original aspect of this project. The accuracy of the sensors was not reported. In contrast, the authors of [43] presented the ENVIRA system [44]. This multi-sensing unit is equipped with air temperature, globe temperature, relative humidity, CO_2_, airspeed, tVOC, pressure, sound pressure, illuminance, and PM_2.5_ sensors. The system was tested in different offices and educational buildings, and the measurement results were employed to provide an indicator of the overall quality of the environment. In this case, the sensors’ accuracy was derived either from the literature or from direct comparisons with reference instruments. In [44], a comprehensive assessment of LCSs for indoor air quality monitoring was carried out through the application of different calibration models. In particular, the performances of CO, NO_2_, and O_3_ sensors were evaluated against different machine learning (ML) methods. We mention this study as it uses the open-source approach and can serve as a starting reference, together with [45], for state-of-the-art artificial intelligence (AI) calibration methods.

#### EQ-OX Concept

These works influenced the authors of the current paper in the design and development of a low-cost, customizable, and inherently flexible platform, Environmental Quality bOX (EQ-OX), to assess the majority of IEQ parameters that are important for building users at the same time. EQ-OX has some main features that differentiate it from previously developed monitoring kits, listed as follows:**Flexibility**—EQ-OX was conceived to allow for the replacement of sensors following developments of the market toward more reliable and robust LCSs. This feature is derived from the development principle adopted for both hardware and software components. Environmental monitoring campaigns ask for modular platforms, i.e., platforms made up of independent hardware parts (data acquisition and transmission cards and sensors) that can be replaced or expanded with ease depending on the specific research focus, sensor maintenance needs, or sensor upgrade actions.**Number of sensors**—compared to previous works, EQ-OX increases the number of monitored parameters to allow for a more in-depth analysis of IEQ and more possibilities to correlate it with human health and productivity.**Lightweight correction algorithm**—The LCS accuracy provided by manufacturers cannot be accepted outright. Many different techniques can be applied to calibrate LCS sensors against reference instruments. Nowadays, a mainstream approach is to involve ML methods for data processing, which demonstrate impressive results in improving the correlation between LCSs and reference time series. However, this requires a considerable amount of computational effort and computer scientists to properly set up and train the sensor calibration pipelines. Moreover, different sensing principles require different AI methods, making the calibration of a multiparameter sensing kit time and energy consuming. In order to ease the adoption of the EQ-OX concept from a software point of view, a lightweight linear correction algorithm is suggested, which, for most of the LCSs analyzed in this work, is able to bring the accuracy performance to acceptable values.

### 1.2. Aim of the Study

The main goal of this paper is to present the development of EQ-OX, a small and cost-effective system that can detect the main indoor hygrothermal parameters and air quality indicators of an indoor environment. We present the results of a comparison between the behavior of EQ-OX and calibrated professional-grade instruments under controlled laboratory conditions or in real indoor environments. Using a benchmarking process, we found the best trade-off between the accuracy of measurements and the costs, and we were able to decide whether some quantities may be neglected or derived from other measurements without jeopardizing our main goal, i.e., to achieve a suitable level of accuracy with reduced costs for the greatest number of diverse parameters. The present paper aims at: (i) detailing the EQ-OX system architecture, which was worked out based on research requirements for indoor environmental quality monitoring; (ii) providing an algorithm to rapidly estimate the improvement in the LCSs’ accuracy that can be obtained by a linear correction after comparing them with calibrated professional-grade instruments; (iii) reporting the ranking obtained for all 15 LCSs embedded in EQ-OX; and (iv) describing a selection of best practices that may further reduce the inaccuracy of measurements when the system is under operational conditions, thus enhancing the trustworthiness of the acquired data.

## 2. Materials and Methods

### 2.1. Description of the EQ-OX System

#### 2.1.1. Case, Main Hardware, and Firmware

The EQ-OX system is composed of a 20 × 12 × 8 cm PLA 3D-printed box designed for this experiment. Figure 1a shows a rendered image of the system, while a sketch of the positioning of the inner sensors is shown in Figure 1b. Table 1 shows the system’s main components and their costs.

The aim of our design was to obtain a compact and robust device, while obtaining the minimal aspect to simplify installation within residential and commercial buildings.

The design also aimed at full scalability and adaptability to different scenarios. For this purpose, each sensor and component has a dedicated internal or external slot where it can be hosted and, if needed, removed. Customized openings allow for easy access to the power connectors and to the main control board, with the possibility of loading or updating the EQ-OX firmware. An open lattice running all around the sides ensures continuous ventilation, which is needed for monitoring air quality parameters and to prevent internal overheating during operation.

Data acquisition and preprocessing are carried out with a customized electronic control unit based on the Microchip Technology (Chandler, AZ, USA) ATmega2560 microcontroller (the same used on the Arduino Mega). This choice was mainly driven by the open-source culture and user community that flourishes around Arduino, which guarantees a prompt crowd-support when software maintenance is needed. The board was designed to promote the customization of the system, with 29 port buses available (9 0–5 V analog ports, 2 voltage dividers for thermistors, 4 digital high–low ports, 8 I^2^C, 2 UART, 4 SPI). JST connectors mounted directly on the board allow for solid connections and mounting/replacing sensors quickly.

The firmware of the EQ-OX motherboard is based on Arduino programming language and is loaded via serial device through USB (type B connector). As for the physical connections, the main data acquisition software can also be updated to onboard new LCS sensors, leveraging standard libraries. In fact, it is most likely that LCSs using the same kind of connection bus require the same conditioning. Of course, the initialization headers may differ, but this information is always provided by the manufacturer. As an example, the specific I2C address stack and the register vocabulary come with the datasheet, in the case of a digital LCS, or the appropriate sensitivity coefficient comes with a calibration report, if we have an analog LCS. The rationale of the code running the EQ-OX is shown as a flowchart in Figure A26, Appendix A.

#### 2.1.2. Data Transmission

The unit is also provided with a Microchip RN2483 868-MHzcommunication module based on LoRaWAN technology. This wireless protocol is highly convenient for environmental monitoring in which the data throughput is limited and low power consumption is crucial [46,47,48]. The LoRaWAN chip sends the acquired data to an independent gateway, which can also collect data from other EQ-OX sensing units. The gateway acts as an information packet forwarder from the measuring points (houses, factories, offices, etc.) to a database through a mobile internet connection established using a router connected via Ethernet directly to the gateway. The LoRaWAN architecture is typically structured in a star or mesh topology [46], in which measurement nodes like EQ-OX devices lay at the edge of a centralized network. The implementation reported in this work encompasses a Multitech Conduit as an independent gateway, a Mikrotik router, and an InfluxDB database hosted on Eurac Research’s server as a data instance storage. Table 2 shows the exact model of the devices used and their costs.

It is worth mentioning that, at the motherboard’s firmware level, the sensor readout data are encrypted into a simplified JSON data packet (compliant with the limited LoRaWAN data throughput), which is then accepted and reshaped by the gateway to obtain the final HTTP POST request for feeding the sensor time series on the database.

To ensure a negligible information packet loss ratio, each sensor measurement is also stored locally on the microSD card of the main control unit. This allows the use of EQ-OX in stand-alone operations and for low-network-coverage areas, such as remote locations or technical rooms often located in the basement of a building. Moreover, the LoRaWAN protocol, working in the 868 MHz band, ensures a good level of penetration for different materials (brick walls, concrete, trees), i.e., lower losses in the presence of obstacles (see [49]). In addition, the low-power radio module is limited to max. 14 dBm, enhancing the energy efficiency of the whole system.

#### 2.1.3. Power Requirements

The system is powered by a universal AC/DC transformer providing a 7.5 V output. The current applications of EQ-OX can take advantage of a direct connection to the power grid since it allows for the usage of remotely controlled smart plugs to switch the system on, according to one’s requirements, and monitor its power consumption. Since the electrical grid is usually available in typical indoor monitored environments, the power connection does not represent a limitation to EQ-OX’s employment in different scenarios. However, to promote further exploitation of EQ-OX in the field, battery powering is currently being investigated. The electrical consumption of EQ-OX depends on which of the selected sensors are used. For a fully equipped sensing unit (i.e., all the sensors are connected at the same time), the average power consumption is about 2.5 W.

### 2.2. Sensors and Experimental Conditions

#### 2.2.1. Sensors and Reference Instruments

Table 3 details the sensors onboarded on EQ-OX. Each sensing unit of the EQ-OX multi-parameter system has been benchmarked against a dedicated reference instrument. Table 4 summarizes the main features of the high-resolution equipment used in these tests.

#### 2.2.2. Experimental Conditions

This section reports our experimental setup for testing purposes. In particular, we can cluster the sensors by how the tests were carried out, namely in supervised environments or unsupervised environments.


**
Sensors tested in supervised conditions:
**
**Air temperature** is measured through a 10k negative temperature coefficient (NTC) thermistor that ranges from −20 to 50 °C, with tabulated accuracy of ±0.2 °C. This is a cheap, reliable, easy-to-use, and adaptable temperature sensor. The characterization of the temperature sensors was carried out in a climatic chamber. This also allowed us to test the sensor considering the influence of the whole EQ-OX monitoring system (e.g., overheating of internal components and shielding of the sensors). As a reference instrument, an RTD Pt100 1/10 DIN sensor (TC Direct, Torino, Italy) was used. The characterization was carried out in the range of 10–35 °C.**Mean radiant temperature (MRT)** is derived by the measurement of globe temperature performed with a black globe thermometer consisting of a black sphere, with a 10 k NTC thermistor inside [50]. The use of a 40 mm black sphere represents a good compromise between the accuracy of the MRT measurement and the response time [51]. The EQ-OX’s 40 mm self-assembled black sphere globe thermometer was compared with a calibrated 150 mm black globe thermometer (model TP875.1.I from DeltaOhm, Padova, Italy featuring a Pt100 class-A temperature sensor). It should be noted that the certificate, which is issued by the suppliers of this type of instrument, concerns a calibration of the internal temperature sensor but no uncertainty is given on the actual globe thermometer. Both sensors were placed in Eurac Research’s climatic chamber, where the temperatures of each wall could be individually controlled. The devices were positioned in the room’s center and a fan was used to modify the ventilation during the different tests to evaluate the influence of both radiative and convective heat exchange on the globe temperature sensor. The temperatures of the walls were programmed to vary from 12 to 30 °C, resulting in a variation in the globe temperatures between 17 and 22 °C. For the correlation, we used a smaller range, from 18.5 °C to 21.5 °C.**Relative humidity** is detected with an SHT31-D CMOS sensor chip from Sensirion, Stäfa, Switzerland, which declares a relative humidity (RH) measuring range between 0% and 100% RH with an accuracy of ±2% RH. Polymer-based capacitive humidity sensors, such as the SHT31, show good linearity in the humidity range that is relevant in most residential and industrial conditions, i.e., between 20% and 80% RH. Yet, for low humidity values (0–20% RH), this kind of sensor often exhibits highly nonlinear characteristics. Ref. [52] presented the results of some experimental trials of CMOS polymer-based capacitive humidity sensors. The relative humidity sensor was characterized inside a climatic chamber. The characterization was performed in the range of 20–80% RH, which covers the most common values found indoors. The characterization was carried out at two different isotherms, at 10 °C and 30 °C, to evaluate the influence of low or high temperatures on the measurement of the relative humidity. The reference sensor used for this test is an E+E EE060 that was previously calibrated in an accredited calibration laboratory (LAT Center). As a general remark, all humidity sensors usually measure humidity between 20% and 98%, because 100% humidity is water, and below 20% is practically dry air. At high humidity above 85%, the problem with polymer sensors is high hysteresis. Namely, when the humidity is above 85%, the sensor needs a very long time (up to 1 min) to dehumidify and measure the humidity again within the ±2% range. There is, however, a method of measuring humidity using the quartz method with an open condenser, in which this hysteresis is minimal (in the range of less than 1 s), as shown in [53].**Surface temperature** is measured with a Melexis MLX90614ESF-BCI (Ypres, Belgium) infrared thermometer mounted on the tip of a flexible arm. The sensor shows an accuracy of ±0.5 °C in the range of −20–50 °C. The sensor has a 5° cone-shaped field of view (FoV) that determines the relationship between the distance and the area of the walls on which the average temperature is measured. Its flexible support allows the sensor to be pointed towards the object whose surface temperature is to be measured (e.g., radiant ceiling or radiant wall). Tests for the EQ-OX surface temperature sensor were carried out using the same configuration as that for the globe thermometer. By controlling the surface temperatures of each wall separately (ceiling and floor included), it is possible to compare the readings of the EQ-OX surface temperature sensor with 1/10 DIN Pt100 thermometers (TC Direct, Torino, Italy) featuring a metal plate terminal connected to the surfaces. During the tests, we set up a dynamic variation in the temperatures of the different walls to simulate actual non-stationary conditions.



**
Sensors tested in unsupervised conditions:
**
**Pressure** is measured with a Bosch Sensortec BMP388 (Reutlingen, Germany) environmental integrated digital sensor that uses a piezoresistive pressure-sensing element to monitor the air pressure in the range of 900 to 1100 hPa (T = 25–40 °C), with absolute accuracy of ±0.5 hPa, as declared by the manufacturer. The pressure sensor was tested in standard conditions, meaning common values of ambient pressure in indoor applications were used, by comparing data from the Bosch Sensortec BMP388 installed on EQ-OX and the portable Delta OHM HD9408T BARO (Padova, Italy) instrument that was used as reference.**Air velocity** is monitored with a SensoAnemo 5150NSF hot-wire anemometer from Sensor Electronic(Gliwice, Poland), i.e., an omnidirectional air velocity and air temperature sensor, specifically sensitive to the medium–low air velocity, which is mainly relevant in indoor environments. The manufacturer states an operating range of 0.05–5 m/s with an accuracy of ±(0.02 m/s + 1.5% of the reading). The omnidirectional anemometers produced by Sensor Electronic are quite expensive in comparison with the other LCSs installed on EQ-OX. The manufacturer calibrates and applies compensation for the impact of temperature changes on air velocity measurements (air temperature during operation may differ from air temperature during calibration) per single unit, and the compensation and correction coefficients are programmed into embedded EEPROM memory. Due to the lack of a wind tunnel or a sufficiently low turbulent airflow generator, it was not possible to carry out internal tests for comparison with other anemometers. The variability in air motion in open field would not allow us to draw conclusions about the suitability of the Sensor Electronic anemometer compared to that of a reference instrument, even if the instruments are placed close to one another. As no previous tests using this instrument were found in the literature, we relied on the data sheets issued by the manufacturer.**Illuminance** is measured with an AMS Osram AG TSL2561 (Premstätten, Austria) light sensor that detects both infrared and visible light in the range of 0–4000 lux, with two different photodiodes to approximate the response of the human eye. As specified in the datasheet, the performance of TSL2561 was characterized by the manufacturer providing the lux approximation equations, which were integrated into a correction software. However, AMS does not provide any traceable calibration certification or any value of measurement uncertainty. A qualitative analysis of the illuminance sensor performances was carried out by comparison with the LI-COR LI-210R (Lincoln, NE, USA) certified instrument under indoor natural illuminance. The reference device also embeds a photodiode as a sensitive element, centered on the visible light band.**Presence/motion** sensor is used to detect the occupancy of the monitored environment. The Parallax 555-28027 (Rocklin, CA, USA) selected for this purpose, is a passive sensor that measures changes in the infrared energy emitted by surrounding objects. The sensor provides a good estimation of the presence of occupants in a monitored environment, even though other details (such as the number and positions of individuals, as well as their activity) are neglected.**Carbon dioxide** is measured through a CO_2_ meter (Ormond Beach, FL, USA) K30 digital CO_2_ sensor based on non-dispersive infrared technology that calculates the percentage of electromagnetic absorption of a particular wavelength, with ±(30 ppm + 3% of the reading) accuracy in the range of 0–2000 ppm. The CO_2_ K30 sensor integrated into EQ-OX was compared with a TSI 7525 (Shoreview, MN, USA) dual-wavelength NDIR CO_2_ sensor with a calibration certificate from an accredited calibration laboratory (LAT). The instruments were placed close to each other inside an office of about 30 m^2^ occupied by 5 to 15 people a day. Data were acquired for a whole week.**Carbon monoxide, nitrogen dioxide, and ozone** are monitored with three 4-electrode electrochemical sensors: CO-A4, OX-A431, and NO2-A43F. The combination of these sensors takes into account the cross-sensitivity of the O_3_ sensor with NO_2_. The three A4 sensors are connected to the same analog front-end (AFE) circuit board, specifically designed for an easy power supply and value readouts, as well as to mitigate electrical noise issues. To monitor the concentration of carbon monoxide, nitrogen dioxide, and ozone, EQ-OX embedded three Alphasense (Great Notley, Braintree, UK) electrochemical sensors, namely CO-A4, NO2-A4, and O3-A4. In the present study, the Alphasense electrochemical sensors were benchmarked against three high-resolution reference instruments, two Horiba APMA-370 (Kyoto, Japan) instruments for CO and NO_2_, and a model 49i ThermoFisher Scientific (Waltham, MA, USA) instrument for O_3_, respectively. In our case, the LCS and the reference instruments were kept in a constant air volume flux (0.9 L/min), in which a suction pipe delivered outdoor air samples into a cabinet hosting the sensing systems. Moreover, the environmental conditions during the test were steady as the cabinet was provided with cooling feedback, capable of controlling both temperature and relative humidity. Such favorable conditions allowed for long-lasting tests (several weeks of continuous benchmarking).**Particulate matter** is detected with a laser scattering particulate detector: the Alphasense OPC-N3 (Great Notley, Braintree, UK). It uses laser beams to detect particles from 0.35 to 40 µm. Count measurements are converted into mass concentrations of PM_1.0_, PM_2.5_, and PM_10_ using embedded algorithms. The device’s performances have already been evaluated in previous studies, such as the one carried out by the authors of [54]. According to the manufacturer, the device could show cross-sensitivity with water vapor molecules for relative humidity above 95%. Ref. [55] reported high errors for increasing values of relative humidity. In our study, the range of interest for relative humidity was 20–80%, in which the PM sensors should show negligible level of cross-sensitivity with water. Tests for the particulate matter sensors were carried out in the same conditions as those of the Alphasense electrochemical sensors. In this case, the comparison was made between EQ-OX (embedding an Alphasense OPC-N3 PM sensor) and a Thermofisher scientific 5030i SHARP (Synchronized Hybrid Ambient, Real-time Particulate Monitor)particulate monitor instrument. The readout algorithm implemented into the OPC-N3 also considers T and RH as correction factors. Anyway, the indoor conditions of the monitoring station hosting the test are controlled; therefore, no extreme T or RH values are encountered.**Total Volatile Organic Compounds (tVOCs)** are assessed with the PID-AH2 sensor from Alphasense (Great Notley, Braintree, UK), which was also tested by the authors of [56], compared to a professional gas chromatograph. This sensor utilizes ultraviolet light to ionize gas molecules. An electric field attracts ions, generating a current which is proportional to the total concentration of VOC. The Alphasense PID-AH2 used in EQ-OX was compared with an Ion Science (Fowlmere ,UK) TIGER Handheld VOC gas detector, a portable PID instrument with a calibration certificate from an LAT. The instruments were used in the same conditions as those of the CO_2_ sensors, i.e., they were placed close to each other for a whole week inside an office of about 30 square meters used by 5 to 15 people a day.**Formaldehyde** concentration is measured with the SEN0231 HCHO sensor from DFRobot (Shanghai, China), i.e., a formaldehyde electrochemical sensor, which features a breakout board that allows for easy connection, has a small size, and has good resolution (0.01 ppm). Monitoring the presence of HCHO in an indoor environment where several sources of this harmful gas may be present is of paramount importance. Yet, the performances of HCHO are rarely assessed in the scientific literature. Also, no information about its accuracy was provided. The most common problem faced while measuring HCHO concentration is that electrochemical sensors can roughly detect formaldehyde because their readings are affected by the whole concentration of VOC gases. The manufacturer of the SEN0231 HCHO sensor module declares that it can detect and measure formaldehyde concentration by itself, but from our first tests, it seemed to be largely affected by the cross-sensitivities to different compounds and alcohols, among others. The comparison was carried out with an Aeroqual (Avondale, New Zealand) EF formaldehyde sensor, placing EQ-OX and the reference instrument in a sealed box wherein different polluting sources were inserted (e.g., oils, candles, etc.). In all the tests, it seemed that this electrochemical sensor did not respond according to the reference.


### 2.3. Correction Algorithm

This section describes the procedure we adopted to perform the assessment of the sensors hosted by the EQ-OX. Since not all EQ-OX sensors could be tested in supervised/lab conditions, a dedicated methodology was implemented to also exploit the dataset collected during field campaigns.

For all LCS sensors, reference time series are available from comparison tests with high-resolution instruments. Therefore, we implemented a customized algorithm, which assesses the maximum correlation achievable between the LCSs and the reference based on a linear regression. The coefficient of determination (R^2^), the mean average error (MAE), and the root mean squared error (RMSE) were calculated both on raw and corrected time series. 

The core of the procedure consists of filtering the sensors’ time series to extract a subset of intervals for which the trend of the reference (REF) measurements can be considered as stationary. This is particularly important for the sensors tested in unsupervised conditions. Indeed, some sensors are strongly influenced by environmental conditions, such as temperature and relative humidity, such as when they are deployed in the field and the quantities measured are affected by continuous fluctuations. However, from the analysis of the time series, it is possible to pinpoint a number of time intervals in which T and/or RH variations remain within a limited range. So, before applying any linear regression, two-step filtering is applied to the time series to only select intervals of measurements that are not affected by abrupt transients of the monitoring parameters and/or the environmental conditions (e.g., T and RH).

To better clarify our assessment strategy, an example is reported hereafter which explains all the steps of the algorithm. The quantities in the example are provided in arbitrary units, as synthetic data are used. In Figure 2a, the hypothetical time series from an LCS are shown by the red line, while the blue line represents the reference instrument. At the same time, the temperature variation is also monitored (green line). For our first step, a filtering operation on the environmental conditions (temperature in the example) is performed as shown by the light-green horizontal bar (Figure 2a). This, in turn, provides some intervals (vertical light-red bars) for the LCS and REF time series, wherein the variation in temperature is reduced to within acceptable limits with respect to the impact that the environmental parameter may have on the instruments. This first filtering step results in the five intervals for LCS and REF that are reported in Figure 2b.

At the same time, the REF dataset is searched (Figure 2b) to find the parts in which the time series are not affected by strong fluctuations or deviations (a simple limiting condition on the first derivative magnitude aids the scope). The result is a subset of four intervals wherein the temperature, as well as the REF measurement, can be considered almost stationary (Figure 3a). At this point, both the REF and LCS data can be averaged within all the intervals detected from the previous steps, obtaining the scatterplot presented in Figure 3b.

Now, using the average values for the REF and LCSs within the four subsets, a linear regression operation can be performed (Figure 4a) to extract the slope and intercept coefficients that will be used for the time series correction (Figure 4b) and the estimation of the R^2^ achievable by linear regression.

In this work, as a common condition for the environmental filtering parameters, we used temperature and relative humidity of 23 ± 3 °C and 40 ± 10%, respectively (clearly, this was not the case when temperature and relative humidity were the parameters under evaluation). For the second step of the filtering procedure, the monitored parameter was considered steady whenever the REF measurement fluctuation was within 5% of the averaged value for at least 1 h (only in the case of unsupervised condition tests).

It is worth pointing out that a linear regression can also be performed if we consider the complete dataset. However, the pre-processing steps described above lower the gap between the results and those obtainable from a calibration procedure in a supervised environment. Indeed, the filtering has a minor effect on the time series collected during the tests in supervised environment, while it makes a remarkable difference to the time series collected during the field campaigns.

In Section 3, the impact of the pre-processing steps is reported for all the EQ-OX sensors, together with the results of the linear regression between the LCSs and the reference instruments’ filtered time series.

## 3. Results

For comparing the LCSs onboarded on EQ-OX, we classified each of them according to the coefficient of determination (R^2^) obtained from the correction algorithm. We defined a scale ranging from one to five, where a score of one means that the maximum value achievable is R^2^ ≤ 0.2 and a score of five means that R^2^ ≥ 0.8. In this, we aim to provide a ranking figure for assessing the LCSs’ quality. The correlation results between the LCSs’ and the reference instruments’ time series before and after the application of the correction algorithm are reported in Table 4. Their corresponding RMSE and MAE values are also listed in the four central columns. The last two columns of Table 5 show the pre- and post-correction quality scores for all of the LCSs. The five circles in a row are filled up according to the scale defined above.

The LCS ranking was accomplished based on the coefficient of determination computed over the full dataset. This fully mimics the procedure that is usually followed in most scientific papers that apply ML to LCSs, in which the dataset coming from a continuous co-location of DUT and REF is split into two parts: the training and the prediction sets [44,45]. In particular, the second column of Table 6 reports the ratio between the number of samples used to calculate the linear regression and the total number of samples belonging to each dataset. We note that, in the case of the tests carried out in an unsupervised environment, the splitting ratio is always less than 10%. This means that the evaluation of the correction is performed for new data, representing mostly transients or at least non-stationary conditions. Moreover, the R^2^ estimation was carried out considering the full range of values collected during the tests, which in most cases was wider than the range of the setpoints used for the linear regression. It is likely that the outermost parts of the full dataset’s range are not identified as stationary setpoints because they result from rapid and steep variations in the measured quantity (pulses, steps, etc.).

Table 6 reports the overall size of the datasets and the subset through which the linear regression is calculated for each LCS. We also list the sensors’ sampling rate (as a pre-processing step, all LCS time series are resampled to the sampling frequency of the reference instruments). Furthermore, statistics such as the average, the minimum, and the maximum values of the full and the filtered REF time series are also listed. The last two columns of the table report the slope and the intercept found for the linear regression related to the various sensors.

For the reader’s convenience, only a few plots regarding the assessments of specific sensors are reported as examples in this section. Figure 5 shows the results obtained for the tVOC sensor. The upper plot represents the linear regression computed on the portion of the dataset filtered against the stationary criteria outlined in Section 2.3. The averaged values are shown with standard deviation error bars. The *x*-axis represents the reference and the *y*-axis represents the device being tested. Both axes share the same scale range and the bisector is shown by the dashed orange line, which serves as a visual guide highlighting how much the LCS diverges from the reference. In the lower plots, a portion of the time series for the LCS and the reference instrument before and after the application of the correction algorithm is shown. A comprehensive collection of all the plots related to the other sensors can be found in Appendix A. For the purpose of experimental replication, a major part of the dataset analyzed in this work is publicly available [57].

We can make a few general remarks starting from what can be observed in Figure 6. It shows the graphical representation of the CO time series pre- and post-correction. The results obtained by our method may seem reasonably accurate, but if we take a closer look at the computed correlation values, low R^2^ scores are found. This can be explained as a visual effect of smoothing (see also Figure A10, Figure A12 and Figure A14). Indeed, the plots clearly show the overlap of the seasonal trends of the LCSs and reference instruments, especially when the time scale covers several days. The steps and pulses of the time series, which distinguish the devices by the different sensing principles, are indeed hidden in the graphical representation. These contributions, however, are the main source of R^2^ degradation.

Indeed, the correction algorithm is not able to consider inertia or intrinsic noises. In particular, it is most likely that one LCS and its reference instrument do not share the same time constants/response and are not affected by the same sources of noise. For instance, most LCSs do react extremely fast to some external stimuli (sometimes even faster than the physical target parameter they are meant to sense) mainly due to readout electronics. In contrast, reference instruments are more accurate by design, and their time series are commonly smoother and better at mimicking the target environmental parameter. More complex AI-based approaches can cope with such problems, at the cost of a greater amount of computational effort and longer calibration sessions [44]. Another factor to consider is the sensing range; for all the tests in which the target environmental parameter was close to the sensitivity of the LCS, low R^2^ values were recorded. For this reason, Table 4’s ranking cannot be generalized for all the possible conditions, but it must be considered as valid only for the reported experimental (indoor) conditions.

In the next session, a detailed analysis of the obtained results is carried out.

## 4. Discussion

### 4.1. LCSs’ Performance Assessment

Here follows a critical discussion of the results for each separate sensor, with references when available to other previous studies, for the sake of comparison. We reiterate that the goal of our work is to characterize the LCSs embedded in EQ-OX (considering various aspects) to obtain to an assessment of their suitability for IEQ monitoring within the ranges of interest. We do not aim to accomplish a full-fledged calibration of the sensors embedded in EQ-OX and therefore deliver a fully validated instrument from a metrological point of view.

**Air Temperature**—The coefficient of determination between the two instruments is close to one even before the linear correction (Figure A1 and Figure A2). Such a satisfactory result already meets the IEQ requirements and the correction does not improve the result significantly.**Relative humidity**—The correlation between the two instruments was high (R^2^ > 0.99), as shown in Figure A3 and Figure A4. At the extremes of this evaluation range, an increased error appeared and reached a maximum of 3% RH. A strongly non-linear behavior for values below 10% RH has also been analyzed by the authors of [52] for polymer-based capacitive humidity sensors, such as the one embedded in EQ-OX.**Globe temperature**—The correlation between the reference and EQ-OX sensor is high (R^2^ > 0.97), and differences of less than 0.5 °C were found throughout the range of interest. However, the curves show divergent behavior between the two devices as the temperature decreases (Figure A5 and Figure A6), with the maximum error at the lower end of the test range. It is, therefore, necessary to further analyze in future studies the behavior of the sensor in a wider operating range by integrating the comparisons with the analytical measurement of the MRT obtained from the walls’ temperature.**Surface temperature**—The dynamic variation in the temperatures of the different walls within the 1 h period to simulate actual non-stationary conditions led to a difference between the EQ-OX measurements, which were affected by the whole field of view exposed to the sensor and the actual surface temperature. This difference is reduced in the intermediate values and increases as the temperature approaches higher or lower values (Figure A7 and Figure A8). Even if more tests are necessary, thanks to our linear correction algorithm, it was possible to increase the R^2^ from 0.72 to 0.96. For this sensor, the error varies with the temperature and shows a maximum value of around ±2 °C for the higher and lower temperatures, while for the intermediate values (i.e., around 22 °C), the error is negligible.**Pressure**—An offset of about 50 hPa was detected, which is much greater than the one declared by the manufacturer as its accuracy (Figure A9 and Figure A10). The application of a linear correction allowed us to increase the R^2^ up to a value of 0.98.**Illuminance**—The illuminance LCS showed a small offset compared to that of the reference instrument (lower than 1% of the LCS’s average values) for all the values except for a few sharp peaks that the LCS is not able to detect properly (with an error rate up to 20%). Yet, the coefficient of determination was improved to 0.80, as shown in Figure A11 and Figure A12.**Carbon dioxide**—For the carbon dioxide sensor, the linear correction performs well as also reported by the authors of [58]. However, the maximum differences between the LCS and REF of about 9% of the average values were found to be three times higher than the ones declared in the datasheet of the manufacturer. The application of the linear correction algorithm led to R^2^ = 0.92 (Figure A13 and Figure A14).**Carbon monoxide, nitrogen dioxide, and ozone**—The average results are reported in Figure A8, Figure A9 and Figure A10. Despite a Pt-1000 temperature detector present on the analog front-end board, no temperature correction was implemented for EQ-OX. The operating conditions will most likely be a standard temperature and pressure; thus, the correction would have been negligible. Other studies have assessed the dependence of the sensors’ readouts on air temperature and humidity. Ref. [59] reported that the CO-B4 sensor was unaffected by humidity and temperature changes during chamber testing. Ref. [60] reported that the EC sensor’s response varied with humidity by a few percentage points, so it had a negligible quantity compared to typical ambient concentrations.

Among the three sensors, O_3_ increases much more from the applied linear correction, going from an initial R^2^ of −2.85 to an R^2^ score of 0.92 after the correction. NO_2_, in contrast, reaches an R^2^ of only 0.68 post-correction, starting from an initial score of 0.47. It is worth highlighting that the pollutants’ concentrations in the air during the tests were very low (a few tens of ppb), as is commonly expected in an indoor environment. Different studies of CO-A4 laboratory and field tests have been carried out by other research centers in the past, including [59,61,62], which obtained excellent results in chamber conditions, with reported R^2^ values greater than 0.99 with respect to those of the reference. However, these field investigations demonstrated a significant deterioration and broad distribution of the sensors’ performances. In our case, we report an increase from 0.14 to 0.79 R^2^ by applying our linear correction.

Similar behavior to the CO sensor was found by the authors of [61] for the NO_2_ Alphasense sensor. The R^2^ was high under laboratory testing conditions and low for outdoor applications, especially in moderate traffic conditions, when the pollutant concentrations were lower.

As highlighted in the report “Summary of air quality sensors and recommendations for application”, there is a very limited amount of scientific literature on electrochemical ozone sensors, which makes their assessment extremely difficult. Using an Alphasense OX-B421 sensor, the authors of [62] studied its performance under chamber and field conditions.

During chamber testing, the correlation between the sensor response and the reference instrument was found to be excellent with an R^2^ value greater than 0.99. However, during testing in the field as part of an air-quality monitoring station in Norway, they found R^2^ values = 0.01–0.66.

**Particulate matter**—OPC-N3 tends to overestimate larger-diameter particles with respect to the reference instrument. This is possibly due to the implementation of the particle classification in 24 range bins. Such pre-processing starts clustering the particle diameters from 0.35 to 2.5 µm with increasing efficiency from about 80% to 101% [63,64]. The best agreement was found for diameters of 1 µm with a post-correction correlation of 0.64. This also resulted in the lowest RMSE and MAE post-correction values: 2.5 and 1.78, respectively.In contrast, the RMSE and MAE values for 2.5 and 10 µm diameter PMs are bigger, confirming that the reference and LCS time series share a lower correlation even though the R^2^ improves after the application of the correction algorithm.**Total Volatile Organic Compounds**—The trend of the tVOC sensor integrated into EQ-OX is similar to that of the reference. However, there is an overestimation of the peaks and general offset between the signals of the two instruments. Figure 5a shows the average values used to find the correction coefficient. The application of the correction algorithm led to an R^2^ = 0.57. In Figure 5b, a portion of the measurement is shown, in which it is possible to see how the two signals almost overlap after the correction. This difference between the LCS and the gas detector may also be due to the sensitivity of the instrument to VOCs that are different from isobutylene, which is the sole gas used for the standard calibration and for the estimation of the response factor. This parameter relates the PID sensor response against a particular VOC with the PID response against the calibration gas (isobutylene). For instance, if the response factor for a particular VOC is 0.5, the PID response is twice that for isobutylene at the same concentration. This, however, works if we know which gas is present within the measurement site. For this reason, the use of PID sensors is usually more often indicated to control the overall trend, gather information on overall air changes, or estimate total concentration peaks. To gain a better understanding of the composition of tVOCs, a gas chromatograph able to identify specific molecules should be used. The limit of such chromatographs is that they cannot be operated continuously; therefore, portable instruments are relevant for the above-mentioned qualitative purposes. The readout of PID-based tVOC sensors is usually biased by the environmental temperature [23], and the producers provide correction formulas and/or tables to cope with this. However, the EQ-OX suite is mostly intended for indoor campaigns; therefore, we did not implement a strategy to implement for significant temperature variations, as they are not expected in operational conditions.**Formaldehyde**—At the moment, we have not found on the market an LCS that can be integrated into the EQ-OX system to monitor the concentration of formaldehyde with reasonable accuracy. For instance, in our tests, the behavior of the LCS with respect to the reference instrument happens to be completely uncorrelated (see Table 5). Various studies on prototypes, such as a formaldehyde sensor based on Cu-codoped ZnO nanomaterial [65] or one that uses UV light to activate a TiO_2_ plate for sensing formaldehyde [66], are showing promising results, but these solutions are not yet available in the market. Due to the crucial importance of the quantification of formaldehyde indoors, developments in this regard will be continuously monitored to find a solution that better fits our needs.

### 4.2. Field Application

The EQ-OX environmental monitoring system has been tested and developed in the framework of different research projects. Some prototypal units have been exploited to collect IEQ data, like hygrothermal parameters and air pollutant concentrations in offices, laboratories, residential buildings, and educational buildings. From these preliminary tests, we suggest some best practices regarding system deployment in order to mitigate possible systematic uncertainties arising from suboptimal operative conditions. In particular, when the aim of a field study is the evaluation of the IEQ experienced by users, the sensing system should be located as close as possible to the position of the occupants in the room. The concentration of different pollutants, as well as the values of temperature, relative humidity, air speed, and so on, may not be uniform in the room. Thus, all the following precautions should be taken.

The device must be placed inside the room in such a way that the measurements are most representative of the usual position of the occupants. In the case of seated people, the ASHRAE Standard 55-2010 [67] recommends 0.6 m as the positioning height of the device for a correct evaluation of air temperature, air speed, and PMV-PPD, while to assess standing occupants’ perception, a height of 1.1 m is preferred.To prevent errors in thermal comfort analyses, the globe thermometer should receive radiative heat from every surface in the room, including the floor. The ideal position is the center of the room.Since, in many real cases, the center of the room may not be available, it is crucial to avoid hidden or closed-off places that may not be representative of the whole occupied volume of the room.For the same reason, also concerning the surface temperature measurements, EQ-OX should be placed far from any surface that presents a different temperature compared to that of the others in the room.Avoid exposing the sensors to direct solar radiation, which can significantly modify the measured temperatures and the behavior of temperature-sensitive sensors (like gas sensors). In the presence of an air conditioning system, avoid placing the device in proximity to the inlet vents since the measured values would not be representative of the room.Avoid conditions that can prevent the air from flowing into EQ-OX, as this could be detrimental to the optimal detection of harmful pollutants.The accuracy of the illuminance sensor is irrelevant compared to the impact of incorrect positioning. The key is to position the device in a manner that ensures that the measurements represent the average environmental conditions.The pollutant sensors have a limited life if subjected to standard environmental conditions. In the case of a system installed in harsh environments, this time may be greatly reduced. Therefore, it is needed to monitor the sensors over time in order to correctly assess their aging.

These general guidelines must be adapted to each specific scenario.

### 4.3. EQ-OX System Costs

Considering the prices of the individual EQ-OX components, which are reported in Table 1 and Table 2, the overall cost of the kit updated in 2021 is between EUR 1500 and 1800 for the version without an anemometer. The cost of the latter depends on the selected sensor and the accuracy required. The one installed in the presented kit is EUR 1000.

In parallel to the hardware costs, a physical maintenance plan to periodically scrub the sensors operating in a dirty environment must be put in place. However, different sensing principles require different cleaning procedures. As an example, solvents, solutions, or ultrasound can be used for electrochemical sensors; special care should be taken not to impair their sensing capabilities by using overly aggressive products. Compressed air or acetone-based lens cleansers can be used for optical sensors. Still, delicate detergent solutions can be used for the external cleaning of most kinds of LCSs. These actions can definitely extend the sensors’ lifetime. However, a careful cost–benefit evaluation has to be carried out, as many of the cleaning and re-calibration procedures that are usually performed for high-accuracy instruments are unaffordable for LCSs. This aspect will be studied extensively in a future work.

In Figure 7, an example of the installation and data collected from a field study using EQ-OX is shown.

### 4.4. Limitations and Perspectives

Our design and study of this fairly complex monitoring device had the following limitations:The occupancy sensor is placed on the top part of the device shell. While this position is convenient from a design point of view, it may cause some issues for the proper detection of occupancy. A new version that is under development will integrate a second sensor on the side of the box.The light sensor, which is placed on the top part of the device, may be shaded by the globe thermometer, the anemometer, or other objects in the room near EQ-OX. While it is difficult to solve this issue from a technical point of view without compromising the compact design of the tool, this can be mitigated with some extra care in one’s selection of where to place the box and its orientation with respect to windows or lights.To make the device as robust as possible, the length of the small metal pole that holds the globe thermometer is quite limited. This may case some small deviations in the readings due to the view factor between the globe and box itself. A version with a detachable globe thermometer was tested, but this was not ideal for the stability of the system. Anyway, as stated before and as reported in Figure A3, we expect a small amount of deviation due to this.As EQ-OX is dedicated to continuous indoor monitoring, all the sensors have been tested and corrected to value ranges that are typical within buildings (residential and offices). Despite this, the work does not provide a range of ideal measures for all the sensors. To do so, the performed tests should have been carried out for the full range of possible values in the indoor environment for all the sensors. This is out of the scope of this publication.Monitoring kits like EQ-OX, especially for air quality parameters, may be used for the acquisition of trends to perform general IAQ evaluations, to spot cause–effect phenomena, and to eventually implement awareness and early warning systems. Anyway, the accurate measurement of absolute values should be validated by high-resolution instruments just when some threshold limits are reached and spotted by EQ-OX.

## 5. Conclusions

The main purpose of the EQ-OX system is to provide robust data collection in real time from different sensors assessing the main parameters (sound excluded) impacting human perception in an indoor environment, with limited costs. Based on a state-of-the-art analysis and with respect to the IEQ standards, it was possible to conceive a stand-alone device capable of monitoring air temperature, globe temperature, surface temperature, pressure, relative humidity, light intensity, the presence of people, the concentration of several compounds (CO_2_, CO, NO_2_, O_3_, VOC), and the amount and dimensions of particulate matter by implementing only LCSs. The aim of the current work is to verify if, out of any of the tested LCSs, (i) high-quality measurements can be obtained even before the correction, (ii) a lightweight algorithm based on filtering and a linear regression can noticeably improve the sensors’ behavior, and (iii) the correlation between the LCSs and REF instrument within the dataset is too poor to apply any correction.

From the comparison tests performed against the calibrated laboratory equipment, we can observe that, for the main hygrothermal parameters, the coefficient of determination between the LCSs and the reference time series are always close to 1 and the use of a correction algorithm might not even be necessary.

On the other hand, the use of LCSs for other target environmental parameters can lead to a lower accuracy of measurements, if the data are not properly corrected. Even by using only a linear correction strategy, as in the current work, it is possible to noticeably increase the coefficient of determination (R^2^). This is the case, for instance, for the surface temperature, carbon dioxide, ozone, and tVOC LCSs benchmarked.

Yet, for some other quantities, namely particulate matter and nitrogen dioxide, the use of the proposed correction algorithm does not significantly increase the coefficient of determination, which is always below 0.5. The reason for this is justified when considering the different sensing principles between the reference and EQ-OX sensors, which are intrinsically non-correlated or cannot be correlated. In the end, the low-cost formaldehyde sensor did not achieve admissible results in this study.

The accessibility of reliable and low-cost field measurements may drive the integration of LCS monitoring systems like EQ-OX in various contexts. As an example, an interesting application is the design of new building management and control systems able to control an indoor environment based on data coming from a distributed and meaningful monitoring system of different parameters at low sensing costs. Also, the paradigm shift described by the authors of [15] (i.e., from standard government-funded air quality monitoring studies performed with certified and expensive instrumentation to widespread, capillary, flexible, and low-cost sensors) may be further pushed by sharing sensors’ performance analyses and shared correction algorithms. Furthermore, during post occupancy evaluation (POE) surveys for IEQ assessments, our capability to easily monitor all the parameters affecting the perception of an environment near a single occupant or at least inside a room might promote a more in-depth understanding of the overall comfort of the location under investigation.

## Figures and Tables

**Figure 1 sensors-24-02176-f001:**
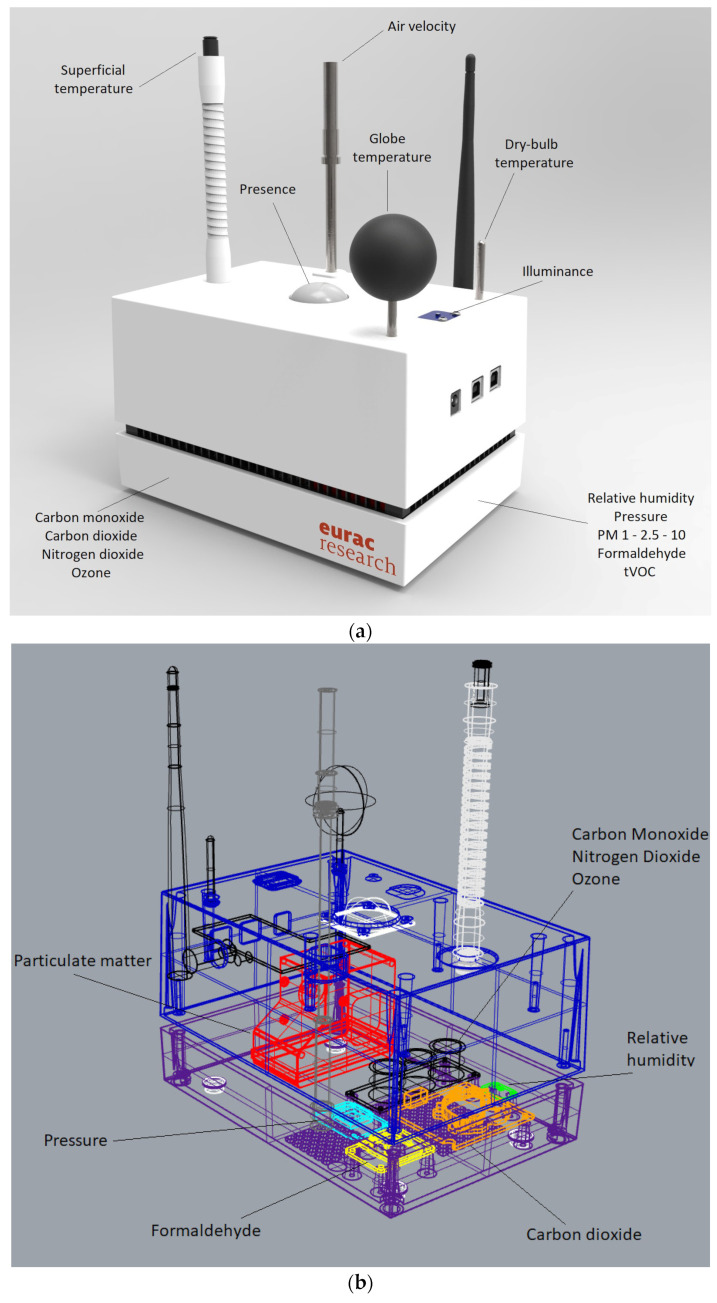
The EQ-OX system: (**a**) a 3D rendering of the system; (**b**) a wireframe view of EQ-OX, highlighting the positioning of the inner sensors (Studio 7B).

**Figure 2 sensors-24-02176-f002:**
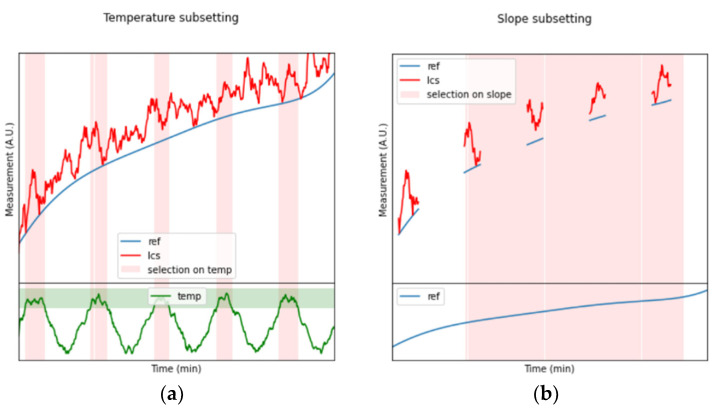
First steps of the LCS time series filtering procedure, in which the variation in the environmental temperature (**a**) and the reference target parameter (**b**) are evaluated to extract only the regions (pink vertical intervals) where the sensor can be considered in steady-state conditions.

**Figure 3 sensors-24-02176-f003:**
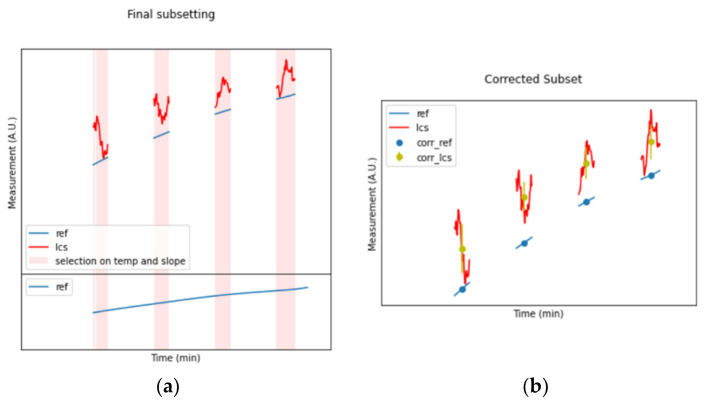
(**a**) Combination of the first two filtering steps, in which only four out of the five pink bars present in (**a**) are kept. (**b**) Averaging applied to the pieces of LCSs and reference time series wherein the environmental conditions are considered stationary.

**Figure 4 sensors-24-02176-f004:**
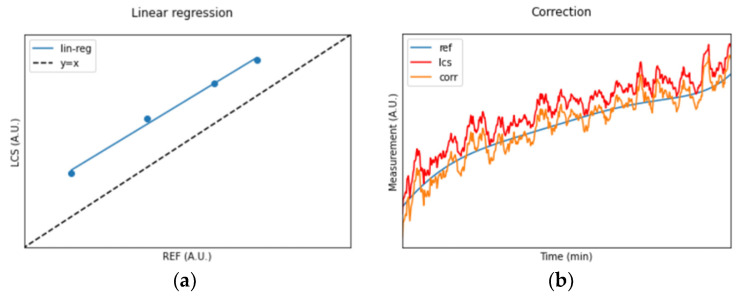
(**a**) Linear regression computed upon the datapoints of Figure 3b. (**b**) Application of the linear regression to the entire LCS time series for comparison with the REF time series after correction.

**Figure 5 sensors-24-02176-f005:**
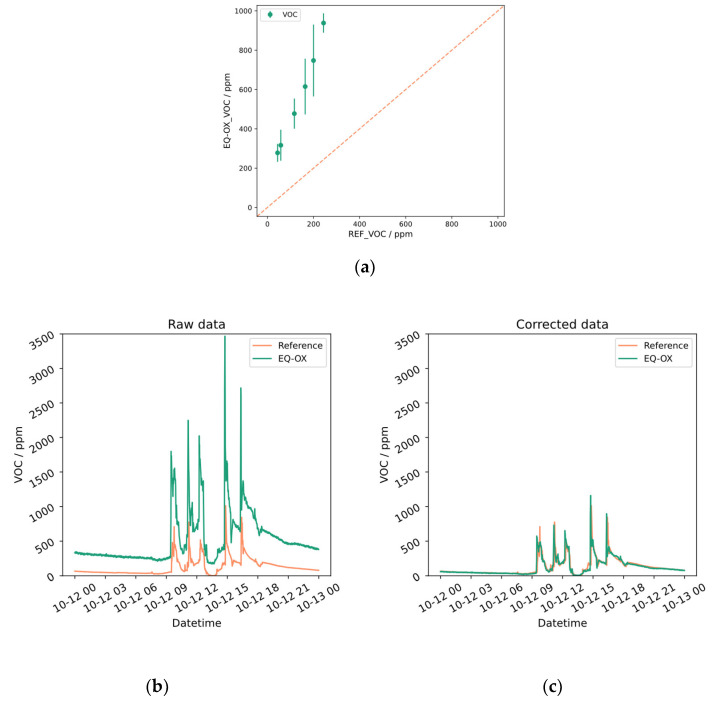
Measurement of the concentration of total VOC (top panel). (**a**) Linear regression plot representing the reference instrument on the *x*-axis and the device being tested on the *y*-axis. Both axes share the same scale range and the bisector is shown by the dashed orange line. (**b**) Portion of data from actual measurement, showing an offset between the reference and EQ-OX. (**c**) Good visual agreement achieved between the two time series after the linear correction algorithm is applied. An R^2^ score from −15.07 to 0.6 confirms the effect of the correction procedure (see Table 4).

**Figure 6 sensors-24-02176-f006:**
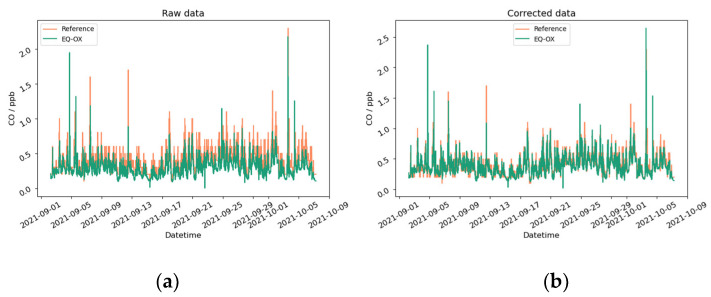
CO concentration acquired from the EQ-OX LCSs and from the reference instrument. (**a**) The uncorrected time series; (**b**) results after the application of the correction algorithm.

**Figure 7 sensors-24-02176-f007:**
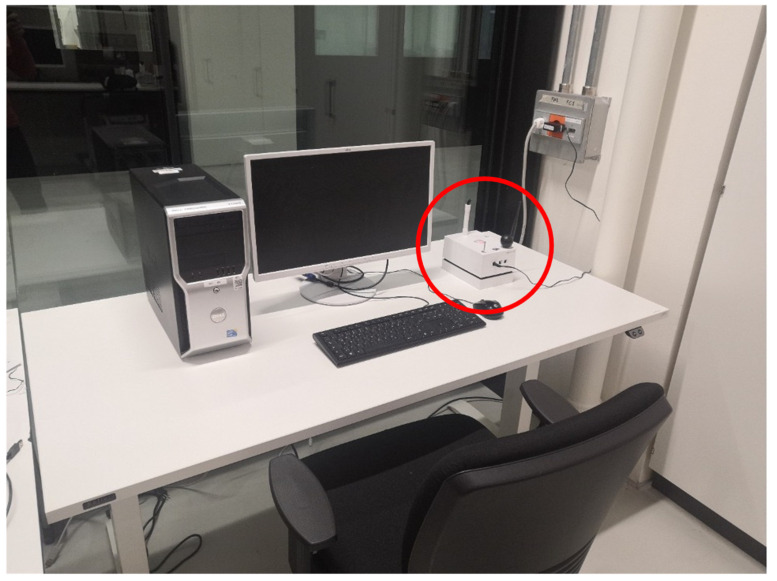
EQ-OX device, highlighted in red circle, during an indoor environmental monitoring campaign.

**Table 1 sensors-24-02176-t001:** General information about the main EQ-OX hardware.

Hardware Component	Brand	Model	Price (EUR)
Motherboard	Eurac Research/Eladit (Bolzano/Pordenone, Italy)	Unique (V.1)	130
Lorawan module	Libelium (Zaragoza, Spain)	LoRaWAN multiprotocol radio shield + Microchip RN2483	110
Custom case	Eurac Research/Studio7B (Bolzano/Brescia, Italy)	Unique (V.1)	50
Power supply	Meanwell (Taipei, Taiwan)	SGAS06E07-P1J	15

**Table 2 sensors-24-02176-t002:** General information about the gateway and router.

Hardware Component	Brand	Model	Price (EUR)
Gateway	Multitech	MTCAP-868-001A	315
Modem/Router	Mikrotik	LtAP Mini LTE	120

**Table 3 sensors-24-02176-t003:** General information about the different LCSs employed for the EQ-OX system. For each measured quantity, the table reports the sensor brand/model, its typology, and price, together with the range, resolution, response time, and accuracy declared by the manufacturer (if available).

Measured Quantity	Brand/ Model	Sensor Type	Range	Resolution	Response Time	Accuracy	Price (EUR)
Dry bulb thermometer	Littlefuse 11492	NTC thermistor	−50 ÷ 150 °C	0.1 °C	<15 s	±0.2 °C	14
Relative humidity	Sensirion SHT31	Capacitive sensor	0 ÷ 100% RH	0.01% RH	<8 s	±2% RH	12
Globe thermometer	Littlefuse 11492	Black globe (40 mm) thermometer	−50 ÷ 150 °C	0.1 °C	<15 s	-	14
Surface thermometer	Melexis MLX90614 ESF-BCI	Infrared thermometer	−40 ÷ 125 °C	0.01 °C	<0.65 s	±0.5 °C	32
Air velocity	Sensor Electronic SensoAnemo 5150NSF	Hot-wire anemometer	0.05 ÷ 5 m/s	0.005 m/s	<1 s	±(0.02 m/s + 1.5% reading)	890
Pressure	Bosch Sensortec BMP388	Piezoresistive sensor	300 ÷ 1000 hPa	0.016 hPa	<0.1 s	±0.5 hPa	15
Illuminance	AMS Osram AG TSL2561	Visible light photodiode	0 ÷ 40,000 lux	1 lux	-	-	16
Presence	Parallax 28027	Pyroelectric sensor	0 ÷ 6 m	−	-	-	12
Carbon dioxide	CO_2_ meter K30	Non-dispersive infrared sensor	0 ÷ 10,000 ppm	1 ppm	<20 s	±(30 ppm + 1.5% reading)	75
Particulate matter	Alphasense OPC-N3	Laser scattering sensor	0 ÷ 2000 μg/m^3^	0.35 μg/m^3^	-	±15% reading	340
Carbon monoxide	Alphasense CO-A4	Electrochemical sensor	0 ÷ 500 ppm	20 ppb	<20 s	-	110
Nitrogen dioxide	Alphasense NO_2-A43F	Electrochemical sensor	0 ÷ 20 ppm	16 ppb	<60 s	-	110
Ozone	Alphasense OX-A431	Electrochemical sensor	0 ÷ 20 ppm	15 ppb	<45 s	-	110
VOC	Alphasense PID-AH2	Photoionization detector	0 ÷ 40 ppm	10 ppb	<3 s	-	400
Formaldehyde	DFRobot Gravity SEN0231	Electrochemical sensor	0 ÷ 5 ppm	10 ppb	<60 s	-	45

**Table 4 sensors-24-02176-t004:** General information about the calibrated instrumentation used to verify the behavior of EQ-OX sensors: for each measured quantity, the table reports instrument brand/model, sensor typology, and price, together with the range, resolution, and accuracy of the measurement declared by the manufacturers.

Measured Quantity	Brand/ Model	Sensor Type	Range	Resolution	Accuracy	Price (EUR)
Dry bulb thermometer	TC Direct 4-wire pt100	RTD pt100 1/10 DIN	−60 ÷ 180 °C	0.01 °C	±0.1 °C	40
Relative humidity	E+E Elektronik EE060	Capacitive sensor	0 ÷ 100% RH	0.01% RH	±3% RH	122
Globe thermometer	Delta Ohm TP875.1.1	Black globe (150 mm) thermometer	−30 ÷ 120 °C	0.01 °C	±0.12 °C	375
Superficial thermometer	TC Direct 4-wire pt100	RTD pt100 Class A	−60 ÷ 180 °C	0.01 °C	±0.3 °C	45
Pressure	Delta Ohm HD9408T Baro	Piezoresistive sensor	800 ÷ 1100 hPa	0.01 hPa	±0.5 hPa	220
Illuminance	LI-COR LI-201R	Visible light photodiode	0 ÷ 100,000 lux	1 lux	±5% reading	720
Carbon dioxide	TSI 7525IAQ—Calc	Non-dispersive infrared sensor	0 ÷ 5000 ppm	1 ppm	±3% reading	2100
Particulate matter	ThermoFischer Scientifc 5030i SHARP	Laser scattering sensor	0 ÷ 10,000 μg/m^3^	0.1 μg/m^3^	±5% reading	3700
Carbon monoxide	Horiba APMA-370	Non-dispersive infrared sensor	0 ÷ 50 ppm	10 ppb	-	3500
Nitrogen dioxide	Horiba APNA-370	Reduced-pressure chemiluminescence sensor	0 ÷ 1 ppm	0.1 ppb	-	3500
Ozone	ThermoFischer Scientifc 49i	UV photometric sensor	0 ÷ 200 ppm	1 ppb	-	3570
VOC	Ion Science Tiger Handheld VOC	Photoionization detector	0 ÷ 20,000 ppm	1 ppb	±5% reading	2875
Formaldehyde	Aeroqual EF HCHO	Electrochemical sensor	0 ÷ 10 ppm	10 ppb	±5% reading	675

**Table 5 sensors-24-02176-t005:** Results of the application of the linear correction algorithm described above. The initial correlation is the value of R^2^ for the LCSs compared with the reference instrument; the final correlation is the value of R^2^ * after the correction has been applied; and the entity of the applied correction can be seen from the values of the slope and intercept of the linear regression curve. The score is based on the value of the R^2^, ranging from 1 for R^2^ < 0.2 to 5 from R^2^ > 0.8.

Measured Quantity	Initial Correlation R^2^	Corrected Correlation R^2^	Initial RMSE	Corrected RMSE	Initial MAE	Corrected MAE	Initial Score	Final Score
Dry bulb thermometer	0.99	1	0.67	0.47	0.62	0.31	◉◉◉◉◉	◉◉◉◉◉
Relative humidity	0.99	0.99	2.08	1.45	1.41	0.43	◉◉◉◉◉	◉◉◉◉◉
Globe thermometer	0.97	0.99	0.16	0.10	0.13	0.08	◉◉◉◉◉	◉◉◉◉◉
Superficial thermometer	0.72	0.96	0.66	0.24	0.55	0.20	◉◉◉◉⚪	◉◉◉◉◉
Pressure	−83.49	0.98	56.40	0.76	56.39	0.61	◉⚪⚪⚪⚪	◉◉◉◉◉
Illuminance	0.61	0.80	716.92	520.10	193.23	106.17	◉◉◉◉⚪	◉◉◉◉◉
Carbon dioxide	0.78	0.92	44.83	27.17	40.12	10.41	◉◉◉◉⚪	◉◉◉◉◉
PM1	−0.21	0.64	4.71	2.50	2.90	1.78	◉⚪⚪⚪⚪	◉◉◉◉⚪
PM 2.5	0.10	0.61	6.15	3.97	3.73	2.87	◉⚪⚪⚪⚪	◉◉◉⚪⚪
PM10	−1.15	0.37	31.98	7.03	15.90	5.16	◉⚪⚪⚪⚪	◉◉⚪⚪⚪
Carbon monoxide	0.14	0.79	0.15	0.07	0.11	0.04	◉⚪⚪⚪⚪	◉◉◉◉⚪
Nitrogen dioxide	0.47	0.68	5.39	3.98	4.24	2.88	◉◉◉⚪⚪	◉◉◉◉⚪
Ozone	−2.85	0.92	29.96	4.20	27.83	3.05	◉⚪⚪⚪⚪	◉◉◉◉◉
VOC	−15.07	0.60	490.52	77.85	397.15	35.15	◉⚪⚪⚪⚪	◉◉◉◉⚪
Formaldehyde	−37.69	0	0.12	0.02	0.12	0.02	◉⚪⚪⚪⚪	◉⚪⚪⚪⚪

* R^2^ can have also negative values. Comparing the fit of the chosen model with that of a horizontal straight line (0 hypothesis), the chosen model could have a worse fit and then R^2^ could be negative.

**Table 6 sensors-24-02176-t006:** Information on the dataset used to perform the linear regression on all LCSs. The ranges and average values are reported together with the slope and intercept obtained from the linear regression calculation. The last entry for formaldehyde is not reported as no linear correction can be calculated on a single datapoint.

Measured Quantity	Regression Dataset Samples	Sampling Rate	Mean REF Total Dataset	Min/Max REF Total Dataset	Mean REF Regression	Min/Max REF Regression	Correction Slope	Correction Intercept
Air temperature	1287 of 2701 tot	10 s	22.9 °C	10.0–35.5 °C	23.2 °C	11.2–35.0 °C	1.01	−0.2
Relative humidity	1079 of 2096 tot	10s	51.0% RH	20.4–97.8% RH	48.7% RH	21.3–77.4% RH	1.08	−3.2
Globe temperature	467 of 1913 tot	1 min	19.7 °C	17.2–21.9 °C	19.9 °C	18.8–21.4 °C	0.93	1.55
Surface temperature	872 of 1921 tot	1 min	19.9 °C	13.3–31.4 °C	20.5 °C	13.8–30.4 °C	1.32	−7.1
Pressure	1192 of 12,245 tot	1 min	990 mbar	974–1000 mbar	986 mbar	975–998 mbar	1.06	3.33
Illuminance	990 of 18,480 tot	1 min	166 lux	0–12,498 lux	227 lux	34–428 lux	1.88	−65.3
Carbon dioxide	1799 of 339,840 tot	1 min	433 ppm	369–1051 ppm	609 ppm	400–936 ppm	1.01	−42.6
PM1	502 of 9755 tot	10 min	7.5 µm/m^3^	0–30 µm/m^3^	9.43 µm/m^3^	3.09–16.67 µm/m^3^	0.56	3.47
PM 2.5	447 of 9755 tot	10 min	10.3 µm/m^3^	0–59 µm/m^3^	19.9 µm/m^3^	3.58–38.0 µm/m^3^	0.72	3.32
PM10	920 of 9755 tot	10 min	16.4 µm/m^3^	0–159 µm/m^3^	19.30 µm/m^3^	7.17–31.0 µm/m^3^	0.27	7.75
Carbon monoxide	518 of 5043 tot	10 min	0.39 ppm	0.1–2.3 ppm	0.41 ppm	0.19–0.60 ppm	1.2	0.02
Nitrogen dioxide	650 of 11,668 tot	10 min	17.38 ppb	2–46 ppb	19.63 ppb	4–37 ppb	1.80	−15.47
Ozone	259 of 11,668 tot	10 min	15.71 ppb	0–74 ppb	15.3 ppb	9–23 ppb	0.61	−11.06
VOC	987 of 220,886 tot	1 min	60 ppb	1–1208 ppb	140 ppb	44–244 ppb	0.25	−6.57
Formaldehyde	202 of 2504 tot	1 min	0.26 ppm	0.18–0.31 ppm	n.d.	n.d.	n.d.	n.d.

## Data Availability

The majority of the dataset analyzed in this work is publicly available [56].

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
