# Peer review of "Environmental Quality bOX (EQ-OX): A Portable Device Embedding Low-Cost Sensors Tailored for Comprehensive Indoor Environmental Quality Monitoring"

_sensors, 2024, doi:10.3390/s24072176_

Round 1

Reviewer 1 Report

Comments and Suggestions for Authors

Comments to the authors:

I suggest some improvement.

Page 2: line 73: The authors should include….. “Electricity consumption is increasing year by year in private blocks in apartments and in public institutions and in data centers as is shown in ref:

-Improved data center energy efficiency and availability with multilayer node event processing. Energies. 2018, vol. 11, no. 9, p. 1-17.  DOI: 10.3390/en11092478

Due to this increasing use of all types of electrical devices, in the field of control and human health, any saving of electrical energy has become a very important feature of the monitoring devices.”

Page 6: line 275, and Table 2, and Table 3: The auhtors should include…. “All humidity sensors usually measure humidity between 20% and 98%. Because 100% humidity is already water, and below 20% is practically dry air. At high humidity above 85%, the problem with polymer sensors is high hysteresis. Namely, when the humidity is above 85%, the sensor needs a very long time (up to 1 min) for the sensor to dehumidify and measure the humidity again within the ± 2% range. There is, however, a method of measuring humidity using the quartz method with an open condenser, where this hysteresis is minimal (in the range of less than 1 second), as shown in the reference:”

-Sensor for high-air-humidity measurement. IEEE T Instrum Meas 1996, 45, (2), 561-563. DOI:  10.1109/19.492787

Page 12: line 443: In addition to the filtering of data from sensors and correction algorithms, the cleaning intervals of the sensors due to a dirty environment are also important. Physical re-calibrations in measurement calibration laboratories authorized for these purposes are also important. Authors should also comment on these procedures.

Reviewer 2 Report

Comments and Suggestions for Authors

The article under review presents a novel device for environmental quality monitoring called EQ-OX. This machine is capable of hosting different kinds of sensors for measuring environmental and air quality parameters, such as, temperature, humidity, air speed, along with carbon dioxide, nitrogen dioxide, carbon monoxide, ozone concentrations, and others. The machine was tested in two different conditions: by placing it in an indoor environment, or in “unsupervised” conditions, as stated by authors; and in “supervised” conditions, which means in a test chamber. The paper surely falls in the focus of the special issue where it has been submitted. Although it is very interesting, in my opinion, it needs to be changed prior potential publication due to the major flaws affecting it. They can be summarized as follows:

11)    The document is very badly structured, and it is very far to be considered a scientific article. As matter of facts, introduction, methods, results and discussion sections are deeply melted and mixed in almost every part of the document. A scientific article must be structured in an introduction, where the authors give all the background and information about previous research, a method (or methodology) section, where the authors just describe the details concerning the materials and the methods needed to accomplish the experiment (or test), a “results” section, where the authors just expose the data concerning the experiment (or test) and/or the numerical indicators related to the performance of the devices under test, a discussion section, where a critical review of the experiment/test is exposed also under the light of previous similar tests/researches, and at last, a conclusion, where final considerations are drawn supported by the results found in the experiment/test. All these elements are deeply melted in the document, creating a huge confusion and very poor readability. For example: table 4, which exposes the results in terms of summary of  the performance indicators, is placed in what can look as the “method/methodology” section, before the “results” section, where it should be shown. Another example: the description of the tests in “supervised” and “unsupervised” conditions, which is clearly an element belonging to the “method/methodology” section, is placed in the “results” section. Moreover, information about previous studies and comparison with them are reported randomly in various parts of the documents, while, the authors should introduce them in the introduction section, and eventually, then dedicate a space in the “discussion” section to discuss and (maybe) make a  comparison with them. Another example  again: in the current version of this manuscript, the “discussion section”  reports just considerations about field applications and costs, and study limitations and its perspectives. This cannot be considered a “discussion” section. The discussion section must show the meaning of the data and of the performance indicators exposed in the “results” section. It must give explanations about poor performance, and eventually, make comparison with results of previous studies. The list of this kind of flaws is very long, therefore, I suggest to rewrite the document under the light of the considerations earlier shortly exposed.

22)    In a scientific article, the authors must provide all the necessary information to replicate the test or the experiment. Concerning this aspect, this document lacks all the information related to the EQ-OX machine, making not replicable the tests exposed in this document. If a reader wants to replicate this experiment, he cannot purchase it (no information about this), nor build a copy of it. Seeing as EQ-OX is not sold, the authors must at least provide all the necessary information to build it. It is fully understandable that, putting all the information related to the hardware and the software of the EQ-OX in this document, is not feasible; therefore, the authors should at least provide links to external sources where the reader can find the required information, and give at least a short, but comprehensive, overview of the internal structure of the machine.

33)    The methods/methodology section lacks important elements. We do not know the number of points on which the linear regressions were performed, or in other words, the number of the records composing the dataset through which the linear regressions are calculated. In an equivalent way, the authors can provide the exact duration of the tests and the sampling rate of the measurements. This aspect is crucial especially in the evaluation of gas and PM sensors. As matter of facts, in general, the bigger is the number of points considered for the linear regression, the better is the coefficient of determination R2. Thus, the authors must insert this information in the document.

44)    It is well known that the PM and the gas sensor performance strongly depends on the levels of concentrations of gases (or air pollutants) present in the test environment. The higher are the concentrations of gases to which they are exposed, the better is their performance. This is particularly true in the case of in-field tests, both in indoor and outdoor environments. Unfortunately, the authors do not provide information about this aspect. They should provide, for example, some statistics about the pollutants monitored by the reference instrumentations, such as, median, mean, min, and max values of the real concentrations of CO, CO2, NO2, VOC, and PM to give a correct idea about the real performance of the devices under test.

55)    The procedure followed to test or evaluate the sensors is not correct. To correctly calibrate sensors or devices, at least two steps (or test periods) are necessary, whatever the environment test is. In the first step the sensor under test is co-located with the reference, thus, the slope and the intercept of the linear regression are found along with the performance indicators (Coefficient of determination, RMSE, MAE). In the second step, usually called “verification” period, the measures of the sensors under test carried out by the linear regression found in the previous step are compared with the reference co-located with the sensors under test. In this way, Coefficient of determination, RMSE, MAE are again computed to give the correct idea about the real performance of the devices under test. The performance indicators related to the “verification” period are usually (and obviously) worse than the ones found in the first period (or step), as it can be found in the dedicated scientific literature. The authors did not follow this scheme, therefore, it is necessary to make again the experiments to correctly evaluate the devices.

66)    By not following the correct procedures to evaluate the sensors, the conclusion drawn by the authors are not correct, or at least, not reliable. Therefore, it should be rewritten after making the tests in the correct way

77)    The “results” section lacks of important data characterizing the test: the slope and the intercept found for the linear regression related to the various sensors.

88)    In the introduction, the panorama related to the previous researches does not consider some previous studies very similar to this one. For example, I suggest to insert in the introduction section a short presentation of study published in the article “Assessment of the Performance of a Low-Cost Air Quality Monitor in an Indoor Environment through Different Calibration Models” (https://doi.org/10.3390/atmos13040567). In this study, an evaluation of some gas sensors was performed through a novel low-cost air quality monitor developed by following an “open-source” approach.

99)    In lines 179-180, the authors claim that the EQ-OX is featured by flexibility. This aspect is very unclear and obscure. The authors claims that it is possible to replace the present set of sensors with others, but they do not explain what kind of sensors are eligible to be used with EQ-OX (are they any type of sensors available on the market? Really??), and more importantly, how. How can the software of the EQ-OX correctly use any sensor connected to its hardware? The author must exhaustively explain this, or at least, avoid to use misleading concepts or ideas.

110) Figures 6a, A1,A3,A5,A7,A8,A10,A12,A14,A16,A18,A20,A22 are really obscure, or at least unuseless to understand the performance of devices. It is not clear at all what x-axes and y-axes represents, as seeing that the plots in the graph are both related to the reference and to the device under test. The author should replace those figures with graphs where the x-axes must represent the reference, and the y-axes must represent the measures of the device under test (the order can be swapped, it’s not important). The plot of the figure must represent the data related to one, or more (if feasible) devices under test.

Reviewer 3 Report

Comments and Suggestions for Authors

sensors- 2850366

Title: A research-grounded tailor-made portable system for indoor environmental quality monitoring implementing low-cost sensors: the Environmental Quality bOX (EQ-OX) system

1.      Unfortunately, I have not found the scientific contribution inside. The novelty of the work should be clearly highlighted.

2.      Aithors just monitored the indoor environmental parameters (air temperature, globe temperature, surface temperature, pressure, relative humidity, light intensity, presence of people, the concentration of several compounds (CO2, CO, NO2, O3, VOC), seem there is n novelty here. Also, the algorithm and the correction methodology implemented are not properly addressed. Calibration of a sensor is fundamental, and here, authors just calibrate the commercially available sensors, which is not novel.

3.      It will be good to add a reference in place of a reference link in the text: “However, even without bringing up the potential health implication due to air contaminants (see, Air quality and health (who.int))”: line 48.

4.      The picture quality is not good. The text inside the figures is non-readable. Please improve.

5.      What is the difference between IEQ and IAQ?

6.      How do authors integrate Arduino to LoRa?

7.  I suggest you expand the mathematical apparatus, describe the measurement methods in detail, etc.

Comments on the Quality of English Language

Minor editing of English language required

Round 2

Reviewer 2 Report

Comments and Suggestions for Authors

dear authors,

thank you for considering my suggestions and modifying the manuscript by following them. The article is hugely improved now. Even though I partially agree with the device calibration issue, I recognize that your work deserves to be published.

Best regards

Reviewer 3 Report

Comments and Suggestions for Authors

Thank you for allowing me to revise the resubmitted manuscript". I believe the submitted manuscript and presented work is suitable for publication in Sensors.